# Predictive Validity of Motor Fitness and Flexibility Tests in Adults and Older Adults: A Systematic Review

**DOI:** 10.3390/jcm11020328

**Published:** 2022-01-10

**Authors:** Nuria Marín-Jiménez, Carolina Cruz-León, Alejandro Perez-Bey, Julio Conde-Caveda, Alberto Grao-Cruces, Virginia A. Aparicio, José Castro-Piñero, Magdalena Cuenca-García

**Affiliations:** 1GALENO Research Group, Department of Physical Education, Faculty of Education Sciences, University of Cadiz, Puerto Real, 11519 Cádiz, Spain; nuria.marin@uca.es (N.M.-J.); carolina.cruz@uca.es (C.C.-L.); julio.conde@uca.es (J.C.-C.); alberto.grao@uca.es (A.G.-C.); jose.castro@uca.es (J.C.-P.); magdalena.cuenca@uca.es (M.C.-G.); 2Instituto de Investigación e Innovación Biomédica de Cádiz (INiBICA), University of Cádiz, 11009 Cádiz, Spain; 3Department of Physiology, Institute of Nutrition and Food Technology “José Mataix Verdú”, University of Granada, 18071 Granada, Spain; virginiaparicio@ugr.es; 4Sport and Health University Research Centre, University of Granada, 18007 Granada, Spain

**Keywords:** speed, agility, prediction, health issues, adults

## Abstract

Motor fitness and flexibility have been linked to several health issues. We aimed to investigate the predictive validity of motor fitness and flexibility tests in relation to health outcomes in adults and older adults. Web of Science and PubMed databases were screened for studies published from inception to November 2020. Two authors systematically searched, evaluated, and extracted data from identified original studies and systematic reviews/meta-analysis. Three levels of evidence were constructed: strong, moderate, and limited/inconclusive evidence. In total, 1182 studies were identified, and 70 studies and 6 systematic reviews/meta-analysis were summarized. Strong evidence indicated that (i) slower gait speed predicts falls and institutionalization/hospitalization in adults over 60 years old, cognitive decline/impairment over 55 years old, mobility disability over 50 years old, disability in instrumental activities of daily living (IADL) over 54 years old, cardiovascular disease risk over 45 years old, and all-cause mortality over 35 years old; (ii) impaired balance predicts falls and disability in IADL/mobility disability in adults over 40 years old and all-cause mortality over 53 years old; (iii) worse timed up&go test (TUG) predicts falls and fear of falling over 40 years old. Evidence supports that slower gait speed, impaired balance, and worse TUG performance are significantly associated with an increased risk of adverse health outcomes in adults.

## 1. Introduction

Increases in life expectancy in addition to the high proportion of deaths being attributed to unspecified causes and the multi-morbidity mainly common in older adults (≥65 years old) are associated with increased rates of healthcare use and cost, resulting in challenges for health policies around the world [1].

Among the dominant causes of death in adults and older adults, non-communicable diseases have a high prevalence and in developing countries; it is estimated that, in coming years, seven out of every ten deaths will be attributed to these diseases [2]. Moreover, disabilities related to pain (such as musculoskeletal pain), depressive disorders, dementia, osteoarthritis, and falls are becoming a great burden in older people [1]. Impaired musculoskeletal health (including mobility and function limitations) is characterized by a reduced quality of life (QoL), loss of independence [3], mental illness, and mortality [4]. Depression is the fourth leading cause of disability and the foremost cause of non-fatal disease burden [5]. In adults and older adults, that interconnection between impaired mental health and QoL is associated with an increased risk of morbidity and mortality [5]. Falls occur in over 30% of older populations [6] and include other adverse outcomes such as lower body fractures, fear of falling, loss of mobility, hospitalization [7], reduced QoL, [8] and even premature death [9]. Collectively, these factors may connive to globally increase healthcare costs [4].

In high-income settings, cardiovascular disease (CVD) is the leading cause of global mortality [10,11,12]. In these countries, a regular medical checkup is well-established in order to control adverse health outcomes, such as CVD risk factors (e.g., hypertension or elevated cholesterol in asymptomatic adults) [13]. Nevertheless, no clear benefits, or even a reduction in adverse health outcomes, such as disability, cognitive decline, risk of falls, well-being, or mortality risk, have been found from annual checkups in the general adult population [13]. In contrast, among older adults, some evidence of a reduction in mortality and an increase in independence has been found [13]. Even so, that examination involves elevated time and health resources and may result in false-positive diagnoses or unnecessary treatment.

It is also well established that by adopting a healthy lifestyle, such as adequate levels of physical fitness, the majority of the aforementioned non-communicable diseases and deterioration in health could be potentially reduced [3,14]. Indeed, maintaining an adequate physical fitness level is considered a powerful marker of health, making it a good indicator of possible adverse health events in different populations including adults and older adults [3,15,16]. In fact, numerous systematic reviews conclude that cardiorespiratory fitness and muscular fitness have predictive validity in relation to diverse health outcomes [10,17,18,19,20,21], being inversely associated with morbidity and all-cause mortality.

On the other hand, there are some other components of physical fitness that need to be analyzed to clarify whether they have predictive value for diverse health outcomes in adults and older adults. Physical fitness is composed of skill-related attributes, such as cardiorespiratory fitness, muscular strength and endurance (muscular fitness), body composition, and flexibility. Motor fitness is considered the performance aspect of physical fitness in daily activities which requires speed of reaction, speed of movement (hereafter, gait speed), agility, coordination, and balance [22]. Flexibility is that component of physical fitness which refers to the ability to move a joint through its full range of motion with ease [23]. These components of physical fitness are related to disabilities that can impair movement common in several health complications, which may imply a reduction in QoL [3]. Therefore, assessment of motor fitness and flexibility could help to identify adults and older adults who may be at risk of suffering from these health outcomes. Through its predictive value, motor fitness and flexibility could be postulated as a health marker in these populations.

Little prior scientific literature has systematically addressed this issue [14,15,24,25,26,27], and there are some limitations and gaps that still need to be investigated thoroughly, since they only were based on older adults, gait speed was the predominant motor fitness test evaluated, and other important health outcomes (such as cause-specific mortality, hip fracture, or mental health and well-being) have not been deeply investigated. Furthermore, no previous systematic review has presented an overview of the different wide range of motor fitness and flexibility tests used in standard health practice (i.e., balance assessment, multidimensional measures, or flexibility assessment, in addition to gait speed assessment). Therefore, the main objective of the present systematic review was to comprehensively analyze the predictive validity of the existing motor fitness and flexibility tests in relation to several health outcomes in adults and older adults. 

## 2. Methods

A systematic review was conducted following the guidelines of the Preferred Reporting Items for Systematic Reviews (PRISMA) [28]. The review was registered in International prospective register of systematic reviews (PROSPERO, registration number: CRD42019140025).

### 2.1. Selected Health Outcomes

The selection of health outcomes was based on recent knowledge of major health problems and their risk factors to seek predictive evidence for associations between motor fitness or flexibility with present and future health status in adults and older adults. The main question was ‘‘does low/high motor fitness or flexibility performance in adults and older adults predict future risk of developing an adverse health status?”

Longitudinal cohort studies and systematic reviews/meta-analysis examining the association between motor fitness and flexibility tests in adults and older adults and future adverse health outcomes were selected: (1) falls and fall-related outcomes (i.e., risk of falls, injurious falls, fear of falling, hip fracture); (2) cognitive impairment (i.e., cognitive decline, dementia, Alzheimer, memory complaints); (3) depressive symptoms (i.e., depression, anxiety, stress); (4) QoL and well-being; (5) disability (i.e., mobility limitations, disability in instrumental activity of daily living, functional decline, dependency, loss of autonomy, frailty, institutionalization or hospitalization risk); (6) pain (i.e., arteriosclerosis, low back pain and/or lumbago); and (7) all-cause mortality, cardiovascular risk and deaths, cancer deaths, and cause-specific deaths.

### 2.2. Search Strategy

An initial search of the electronic databases Medical Literature Analysis and Retrieval System Online (MEDLINE, via PubMed) and Web of Science was performed from date of inception to November 2020, screening for predictive validity of motor fitness and flexibility tests in adults and older adults, using keyword search terms.

When using PubMed, we included medical subject heading (MeSH) terms to enhance the power of the search. MeSH is the National Library of Medicine’s controlled vocabulary thesaurus used for indexing articles for PubMed. The same search strategy and combination of terms was repeated in Web of Science but without using MeSH terms or equivalent since a similar option does not exist in Web of Science.

The keywords used in search strategy were related to the following topics: (1) participants: young adults (19–44 years old), middle-aged (45–64 years old), and elderly (≥65 years old); (2) outcomes: falls, bone health, dementias, depression, anxiety, stress, quality of life, disability, frailty, institutionalization or hospitalization, pain, cardiovascular risk, mortality; (3) exposures: measures of motor fitness and flexibility: gait speed, walking speed, postural balance, agility, range of motion, flexibility; and (4) design: longitudinal studies, prospective study, retrospective study design and prospective follow-up. The four search topics were combined using the Boolean searching with “AND” and “OR” operators. Keywords search terms and search strategies used were different for each database (Appendix A).

### 2.3. Selection Criteria

Two authors working independently (F.M.A. and N.M.J.) read the studies and checked whether they met the inclusion/exclusion criteria. They obtained 93% agreement on the papers selected for the review before consensus, and 100% agreement after discrepancies were resolved in a consensus meeting. If necessary, a third author (JCP) was involved in decision making.

The inclusion criteria for this systematic review was: (1) the study was an original full report published in a peer-reviewed journal, in English or Spanish; (2) the study design was a longitudinal, prospective, retrospective study or prospective follow-up, or systematic review/meta-analysis (if there was more than one study on the same population dataset, the most recent study or the one with the most data best addressing the inclusion criteria was chosen); (3) the study population was a healthy community-based population (only those with cardiovascular risk factors were included), older than 18 years old; (4) one or more motor fitness and flexibility tests were carried out; and (5) the outcome measure was one of the selected health outcomes. Thereafter, if an original study was previously identified through selected systematic reviews/meta-analysis, it was excluded.

When a study title seemed relevant, the abstract was reviewed for eligibility. When more information was required, the full text of the study was retrieved and appraised. The screening of relevant systematic reviews/meta-analysis was also included. In addition, the literature search was complemented by the manual review of reference lists obtained from the selected studies.

### 2.4. Data Extraction

Data were extracted from each study independently by the same two researchers (FMA, NMJ). A preformatted spreadsheet was created for both researchers and included the following columns. For single studies: first author’s last name and year of publication, gender and sample size, mean age of participants and/or age range, length of follow-up (in years), motor fitness or flexibility test assessment, health outcomes, main outcomes and conclusions, and study quality. For systematic reviews/meta-analysis: first author’s last name and year of publication, type of review and number of studies included, age range, motor fitness or flexibility test assessment employed, health outcomes explored, main results and conclusions, and study quality. Any discrepancies in data extraction were discussed until an agreement was made.

A meta-analysis was not completed, due to the heterogeneity (statistical methods, follow-up lengths, protocol, and cut-off points of the motor fitness and flexibility tests, etc.) within the original studies included. Rather, a synthesis of the best available evidence was conducted, examining the methodological quality of each study and the predictive validity of each motor fitness and flexibility tests to identify selected health outcomes. When possible, cut-off points of these tests to identify health outcomes were also included.

### 2.5. Quality Assessment

A quality assessment list for longitudinal studies [29] was used to assess the methodological risk of bias in eligible studies. The list included five items based on population, designs, methods, and report of the results. The items on the list were rated as “1” (positive), “0” (negative) or “?” (unclear), (Table 1). For all studies, a total quality score was calculated by counting up the number of positive items (a total score between 0 and 5). Studies were defined as very low quality if they had a total score less than 2; a total score of 2 was defined as low quality, and a score of 3 or higher was defined as high quality [29].

Potential risk of bias of the selected systematic reviews/meta-analysis was assessed using the assessment of multiple systematic reviews (AMSTAR) rating scale [30]. AMSTAR contains 11 items to appraise the methodological aspects of the reviews: (1) ‘a priori’ design provided; (2) duplicate study selection/data extraction; (3) comprehensive literature search; (4) status of publication as inclusion criteria (i.e., grey or unpublished literature); (5) list of studies included/excluded provided; (6) characteristics of included studies documented; (7) scientific quality assessed and documented; (8) appropriate formulation of conclusions (based on methodological rigor and scientific quality of the studies); (9) appropriate methods of combining studies (homogeneity test, effect model used and sensitivity analysis); and (10) assessment of publication bias (graphic and/or statistical test); and (11) conflict of interest statement. The possible scores were “Yes”, “No”, “Cannot Answer”, or “Not Applicable”. A total possible score of 11 was calculated, counting only for positive responses (“Yes”). The final quality rates were computed by tertiles, where the first tertile ranged from 0 to 3 points, the second tertile from 4 to 7 points, and the third tertile from 8 to 11 points. Likewise, each tertile was treated as low, medium, or high quality, respectively (Table 2).

The same two reviewers (FMA and NMJ) evaluated the quality of the original studies and systematic reviews/meta-analysis separately. A consensus meeting was arranged to sort out differences between both reviewers. They obtained 94% agreement on the quality assessment before the consensus, and 100% agreement after the discrepancies were resolved in a consensus meeting. If there was disagreement, another investigator (JCP) was encouraged to reach consensus. The studies were not blinded for authors, institution, and journal, because the reviewers who performed the quality assessment were familiar with the literature.

**Table 1 jcm-11-00328-t001:** List of included longitudinal studies with quality scores with reference to predictive value of motor fitness and flexibility tests for health outcomes in adults and older adults.

Study	Fitness Components	Non-Selective Population	Clear Health Outcomes	PF and Health Measurement >1 Year	Confounders	SE/CI Information	Total Score
Abu et al., 2018 [31]	Motor fitness (Speed)	1	1	0	0	1	**3**
Brach et al., 2012 [32]	Motor fitness (Speed)	1	1	0	0	1	**3**
Sakurai et al., 2017 [33]	Motor fitness (Speed)	1	1	0	0	1	**3**
Dargent-Molina et al., 1999 [34]	Motor fitness (Speed)	1	1	1	1	0	**4**
Doi et al., 2013 [35]	Motor fitness (Speed)	1	1	0	1	1	**4**
Kang et al., 2017 [36]	Motor fitness (Speed)	1	1	0	1	1	**4**
Kauppi et al., 2014 [37]	Motor fitness (Speed)	1	1	0	1	1	**4**
Laukkanen et al., 2000 [38]	Motor fitness (Speed)	1	1	1	0	1	**4**
Makizako et al., 2015 [39]	Motor fitness (Speed)	1	1	1	0	1	**4**
Abe et al., 2019 [40]	Motor fitness (Speed)	1	1	1	1	1	**5**
Adachi et al., 2019 [41]	Motor fitness (Speed)	1	1	1	1	1	**5**
Andrasfay et al., 2020 [3]	Motor fitness (Speed)	1	1	1	1	1	**5**
Blain et al., 2010 [42]	Motor fitness (Speed)	1	1	1	1	1	**5**
Briggs et al., 2019 [43]	Motor fitness (Speed)	1	1	1	1	1	**5**
Buracchio et al., 2010 [44]	Motor fitness (Speed)	1	1	1	1	1	**5**
Callisaya et al., 2016 [7]	Motor fitness (Speed)	1	1	1	1	1	**5**
Camargo et al., 2016 [45]	Motor fitness (Speed)	1	1	1	1	1	**5**
Deshpande et al., 2013 [46]	Motor fitness (Speed)	1	1	1	1	1	**5**
Doi et al., 2020 [47]	Motor fitness (Speed)	1	1	1	1	1	**5**
Elbaz et al., 2013 [48]	Motor fitness (Speed)	1	1	1	1	1	**5**
Georgiopoulou et al., 2016 [49]	Motor fitness (Speed)	1	1	1	1	1	**5**
Heiland et al., 2018 [11]	Motor fitness (Speed)	1	1	1	1	1	**5**
Hoogendijk et al., 2020 [50]	Motor fitness (Speed)	1	1	1	1	1	**5**
Idland et al., 2013 [51]	Motor fitness (Speed)	1	1	1	1	1	**5**
Jung et al., 2018 [52]	Motor fitness (Speed)	1	1	1	1	1	**5**
Lee et al., 2017 [12]	Motor fitness (Speed)	1	1	1	1	1	**5**
Looker et al., 2015 [53]	Motor fitness (Speed)	1	1	1	1	1	**5**
Luukinen et al., 1995 [54]	Motor fitness (Speed)	1	1	1	1	1	**5**
Makizako et al., 2010 [55]	Motor fitness (Speed)	1	1	1	1	1	**5**
Muraki et al., 2013 [9]	Motor fitness (Speed)	1	1	1	1	1	**5**
Nakamoto et al., 2015 [56]	Motor fitness (Speed)	1	1	1	1	1	**5**
Niiranen et al., 2019 [57]	Motor fitness (Speed)	1	1	1	1	1	**5**
Nofuji et al., 2016 [58]	Motor fitness (Speed)	1	1	1	1	1	**5**
Ojagbemi et al., 2015 [59]	Motor fitness (Speed)	1	1	1	1	1	**5**
Osuka et al., 2020 [60]	Motor fitness (Speed)	1	1	1	1	1	**5**
Quach et al., 2011 [8]	Motor fitness (Speed)	1	1	1	1	1	**5**
Rosso et al., 2019 [61]	Motor fitness (Speed)	1	1	1	1	1	**5**
Sabia et al., 2014 [62]	Motor fitness (Speed)	1	1	1	1	1	**5**
Sanders et al., 2012 [63]	Motor fitness (Speed)	1	1	1	1	1	**5**
Sanders et al., 2016 [64]	Motor fitness (Speed)	1	1	1	1	1	**5**
Stenhagen et al., 2013 [6]	Motor fitness (Speed)	1	1	1	1	1	**5**
Stijntjes et al., 2017 [65]	Motor fitness (Speed)	1	1	1	1	1	**5**
Tian et al., 2019 [66]	Motor fitness (Speed)	1	1	1	1	1	**5**
Veronese et al., 2017 [67]	Motor fitness (Speed)	1	1	1	1	1	**5**
Veronese et al., 2017b [68]	Motor fitness (Speed)	1	1	1	1	1	**5**
Ward et al., 2016 [69]	Motor fitness (Speed)	1	1	1	1	1	**5**
Wihlborg et al., 2015 [70]	Motor fitness (Speed)	1	1	1	1	1	**5**
Zucchelli et al., 2019 [71]	Motor fitness (Speed)	1	1	1	1	1	**5**
Pajala et al., 2008 [72]	Motor fitness (Balance)	1	1	0	0	1	**3**
Austin et al., 2007 [73]	Motor fitness (Balance)	1	1	1	0	1	**4**
Ersoy et al., 2009 [74]	Motor fitness (Balance)	1	1	0	1	1	**4**
Frames et al., 2018 [75]	Motor fitness (Balance)	1	1	1	0	1	**4**
Maki et al., 1994 [76]	Motor fitness (Balance)	1	1	0	1	1	**4**
Muir et al., 2010 [77]	Motor fitness (Balance)	1	1	0	1	1	**4**
Mulasso et al., 2017 [78]	Motor fitness (Balance)	1	1	0	1	1	**4**
Swanenburg et al., 2010 [79]	Motor fitness (Balance)	1	1	0	1	1	**4**
Blain et al., 2010 [42]	Motor fitness (Balance)	1	1	1	1	1	**5**
Breton et al., 2014 [80]	Motor fitness (Balance)	1	1	1	1	1	**5**
Bullain et al., 2016 [81]	Motor fitness (Balance)	1	1	1	1	1	**5**
Cooper et al., 2014 [16]	Motor fitness (Balance)	1	1	1	1	1	**5**
Idland et al., 2013 [51]	Motor fitness (Balance)	1	1	1	1	1	**5**
Kwan et al., 2012 [82]	Motor fitness (Balance)	1	1	1	1	1	**5**
Luukinen et al., 1995 [54]	Motor fitness (Balance)	1	1	1	1	1	**5**
Makizako et al., 2010 [55]	Motor fitness (Balance)	1	1	1	1	1	**5**
Nakamoto et al., 2015 [56]	Motor fitness (Balance)	1	1	1	1	1	**5**
Nitz et al., 2013 [83]	Motor fitness (Balance)	1	1	1	1	1	**5**
Nofuji et al., 2016 [58]	Motor fitness (Balance)	1	1	1	1	1	**5**
Vellas et al., 1997 [84]	Motor fitness (Balance)	1	1	1	1	1	**5**
Wihlborg et al., 2015 [70]	Motor fitness (Balance)	1	1	1	1	1	**5**
Abu et al., 2018 [31]	Motor fitness (Speed–agility)	1	1	0	0	1	**3**
Sakurai et al., 2017 [33]	Motor fitness (Speed–agility)	1	1	0	0	1	**3**
Austin et al., 2007 [73]	Motor fitness (Speed–agility)	1	1	1	0	1	**4**
Clemson et al., 2015 [85]	Motor fitness (Speed–agility)	1	1	1	0	1	**4**
Davis et al., 2015 [86]	Motor fitness (Speed–agility)	1	1	0	1	1	**4**
Doi et al., 2013 [35]	Motor fitness (Speed–agility)	1	1	0	1	1	**4**
Ersoy et al., 2009 [74]	Motor fitness (Speed–agility)	1	1	0	1	1	**4**
Kang et al., 2017 [36]	Motor fitness (Speed–agility)	1	1	0	1	1	**4**
Mulasso et al., 2017 [78]	Motor fitness (Speed–agility)	1	1	0	1	1	**4**
Asai et al., 2020 [87]	Motor fitness (Speed–agility)	1	1	1	1	1	**5**
Breton et al., 2014 [80]	Motor fitness (Speed–agility)	1	1	1	1	1	**5**
Doi et al., 2019 [88]	Motor fitness (Speed–agility)	1	1	1	1	1	**5**
Donoghue et al., 2017 [89]	Motor fitness (Speed–agility)	1	1	1	1	1	**5**
Kwan et al., 2012 [82]	Motor fitness (Speed–agility)	1	1	1	1	1	**5**
Nitz et al., 2013 [83]	Motor fitness (Speed–agility)	1	1	1	1	1	**5**
Savva et al., 2013 [90]	Motor fitness (Speed–agility)	1	1	1	1	1	**5**
Schroll et al., 1997 [91]	Motor fitness (Speed–agility)	1	1	1	1	1	**5**
Bravell et al., 2017 [92]	Flexibility	1	1	1	1	1	**5**
Luukinen et al., 1995 [54]	Flexibility	1	1	1	1	1	**5**
Ward et al., 2016 [69]	Flexibility	1	1	1	1	1	**5**

PF, Physical Fitness component; SE, Standard Error; CI, Confidence Intervals. Bold values are defined as total score.

**Table 2 jcm-11-00328-t002:** Quality assessment of the systematic reviews included using the AMSTAR ^#^ rating tool.

Reviews	1	2	3	4	5	6	7	8	9	10	11	Rating	Quality *
Van Kan et al., (2009) [15]	Yes	Yes	No	No	No	Yes	Yes	Yes	No	N/A	Yes	6	Medium
Grande et al., (2019) [26]	Yes	Yes	Yes	Yes	No	Yes	Yes	Yes	No	N/A	Yes	8	High
Cavanaugh et al., (2018) [14]	Yes	Yes	Yes	Yes	No	Yes	Yes	Yes	Yes	N/A	Yes	9	High
Wang et al., (2020) [27]	Yes	Yes	Yes	No	No	Yes	Yes	Yes	Yes	Yes	Yes	9	High
Quan et al., (2017) [24]	Yes	Yes	Yes	Yes	No	Yes	Yes	Yes	Yes	Yes	Yes	10	High
Peel et al., (2019) [25]	Yes	Yes	Yes	Yes	No	Yes	Yes	Yes	Yes	Yes	Yes	10	High

^#^, AMSTAR contains 11 items to appraise the methodological aspects of the systematic reviews. All 11 items were scored as “Yes”, “No”, “Can’t Answer” (C/A), or “Not Applicable” (N/A). A total possible score of 11 was calculated, counting only for positive responses (“Yes”). Quality *, The final quality rates were computed by tertiles, where the first tertile ranged from 0 to 3 points, the second tertile from 4 to 7 points, and the third tertile from 8 to 11 points. Likewise, each tertile was treated as “low”, “medium”, or “high” quality, respectively. ^#^, All 11 items were scored as “Yes”, “No”, “C/A”, or “N/A”. AMSTAR comprises the following items: 1. “a priori” design provided; 2. duplicate study selection/data extraction; 3. comprehensive literature search; 4. status of publication as inclusion criteria (i.e., grey or unpublished literature); 5. list of studies included/excluded provided; 6. characteristics of included studies documented; 7. scientific quality assessed and documented; 8. appropriate formulation of conclusions (based on methodological rigor and scientific quality of the studies); 9. appropriate methods of combining studies (homogeneity test, effect model used and sensitivity analysis); 10. assessment of publication bias (graphic and/or statistical test); and 11. conflict of interest statement.

### 2.6. Levels of Evidence

Three levels of evidence were constructed [93]: (1) strong evidence: consistent findings in three or more high-quality studies; (2) moderate evidence: consistent findings in two high-quality studies; (3) limited or inconclusive evidence: consistent findings in multiple low-quality studies, inconsistent results found in multiple high-quality studies, or results based on one single study.

## 3. Results4

### 3.1. Study Selection

Figure 1 presents the flow-chart of retrieved and selected studies and systematic reviews/meta-analysis. The electronic search strategy retrieved 1274 studies, from PubMed and Web of Science databases. Duplicate references were removed, resulting in 1102 studies screened. Of these, 929 records were excluded by not meeting tittle and abstract criteria, while 173 full-text studies were assessed for eligibility. Thereafter, 95 studies and 2 systematic reviews were further excluded for different reasons. The main reasons for rejection were: unable to source full text; no predictive validity; no language inclusion criteria; no healthy population; no fitness test assessed; and no complying with study designs criteria; original studies previously identified through selected systematic reviews/meta-analysis. A list of these excluded studies (*n* = 1026) is provided in Appendix A. Finally, 70 studies met the inclusion criteria and were included in the systematic review, 6 of them as systematic reviews/meta-analysis.

### 3.2. Risk of Bias within Studies

Table 1 shows the list of included longitudinal studies with quality scores. Overall, methodological quality was considered high, without any study scoring 2 or less, that is, with a low or very low quality. The quality assessment of included systematic reviews/meta-analysis is presented in Table 2. Five of them were classified as high quality [14,24,25,26,27] and the remaining one as medium quality [15].

### 3.3. Characteristic of Included Studies

The overall number of participants in the selected studies [3,6,7,8,9,11,12,16,32,33,34,35,36,37,38,39,40,41,42,43,44,45,46,47,48,49,50,51,52,53,54,55,56,57,58,59,60,61,62,63,64,65,66,67,68,69,70,71,72,73,74,75,76,77,78,79,80,81,82,83,84,85,86,87,88,89,90,91,92,93] was 100,422 (51.5% females), while in five studies, sex was not specified [43,46,58,59,62]. The ages of the participants ranged from 35 to 105 years, with a mean age of 72. The length of follow-up ranged from 0.5 to 25 years (detailed characteristics and findings of the selected studies in Appendix A).

A total of three systematic reviews [14,15,26] and three meta-analyses [24,25,27] were included. The overall sample size involved 274,871 participants. Specific sex was not usually reported. The ages of the participants ranged from 54 to 108 years old. The follow-up period ranged from 11 days to 25 years (detailed characteristics and findings of the selected systematic reviews/meta-analysis in Appendix A).

Table 3 and Appendix A summarize the main characteristics, motor fitness and flexibility tests and health outcomes of the selected longitudinal studies and systematic reviews/meta-analysis, as well as a broad picture of significant/non-significant associations found between motor fitness and flexibility tests with health outcomes. Appendix A presents the number of studies reporting predictive validity or lack of predictive validity of the included motor fitness and flexibility tests for different health outcomes. The highest number of predictive results were found for gait speed, followed by postural balance assessment and the multidimensional measurement test, timed up&go test (hereafter, TUG test).

**Table 3 jcm-11-00328-t003:** Main characteristics and findings of longitudinal studies investigating the predictive validity of motor fitness and flexibility tests for health outcomes in adults and older adults.

Author	N (% Female Sex)	Mean (Range) Age	Fitness Test	Health Outcomes
Gait Speed	Balance	TUG	Flexibility	Other	Falls/Hip fracture	Cognitive Decline/Impairment	Depression/Well-Being	Related-Mobility Disability/ Frailty	CVD/Stroke	All-Cause Mortality	Other-Causes Mortality
Kang et al. 2017 [36]	541 (57)	67 (60–86)						 						
Quach et al., 2011 [8]	763 (64)	78												
Callisaya et al., 2016 [7]	509 (52)	75 (60–105)												
Abu et al., 2018 [31]	325 (55)	68 (60–89)						 						
Kwan et al., 2012 [82]	280 (43)	75 (65–91)						 						
Muraki et al., 2013 [9]	2215 (66)	68												
Sanders et al., 2016 [64]	4112 (53)	74 (≥65)	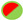											
Luukinen et al., 1995 [54]	1016 (63)	76 (70–92)						 						
Doi et al., 2013 [35]	57 (74)	80 (≥65)												
Stenhagen et al., 2013 [6]	1763 (48)	76 (60–93)												
Dargent-Molina et al., 1999 [34]	5895 (100)	80 (≥75)												
Kauppi et al., 2014 [37]	2300 (58)	66 (≥55)												
Wihlborg et al., 2015 [70]	1044 (100)	75												
Ersoy et al., 2009 [74]	125 (100)	61 (50–79)												
Muir et al., 2010 [77]	90 (37)	80 (60–90)												
Austin et al., 2007 [73]	1282 (100)	75 (70–85)												
Vellas et al., 1997 [84]	267 (58)	73 (≥60)												
Mulasso et al., 2017 [78]	19 (62)	73 (≥65)												
Nitz et al., 2013 [83]	449 (100)	59 (40–80)												
Swanenburg et al., 2010 [79]	270 (83)	73 (60–90)												
Maki et al., 1994 [76]	100 (83)	83 (62–96)												
Pajala et al., 2008 [72]	434 (100)	70 (63–76)												
Frames et al., 2018 [75]	98 (66)	75												
Asai et al., 2020 [87]	649 (65)	76 (≥60)												
Clemson et al., 2015 [85]	1000 (53)	73 (65–94)												
Doi et al., 2020 [47]	3696 (53)	71 (≥65)												
Makizako et al., 2015 [39]	948 (49)	78 (≥75)												
Heiland et al., 2018 [11]	1756 (66)	71 (≥60)										* 		
Abe et al., 2019 [40]	973 (48)	75 (≥65)												
Makizako et al., 2010 [55]	265 (55)	79 (68–96)									*  			
Adachi et al., 2019 [41]	516 (81)	79 (76–82)												
Laukkanen et al., 2000 [38]	388 (67)	77 (75–80)												
Nakamoto et al., 2015 [56]	961 (48)	60 (40–79)		* 										
Brach et al., 2012 [32]	552 (61)	79 (≥65)												
Deshpande et al., 2013 [46]	622 (?)	67 (50–85)												
Rosso et al., 2019 [61]	337 (51)	78 (70–79)												
Jung et al., 2018 [52]	1348 (55)	76 (≥65)												
Breton et al., 2014 [80]	1265 (52)	73 (68–82)												
Ward et al., 2016 [69]	391 (67)	77 (≥65)												
Savva et al., 2013 [90]	1814 (51)	70 (≥65)												
Schroll et al., 1997 [91]	259 (56)	77 (75–80)									* 			
Elbaz et al., 2013 [48]	6267 (29)	45 (35–55)												
Zucchelli et al., 2019 [71]	3363 (65)	75 (≥60)												
Andrasfay et al., 2020 [3]	887 (48)	70 (≥60)												
Niiranen et al., 2019 [57]	3453 (54)	55 (45–74)												
Hoogendijk et al., 2020 [50]	4220 (53)	72 (≥55)												
Nofuji et al., 2016 [58]	1085 (?)	77 (65–89)												
Lee et al., 2017 [12]	911 (45)	65												
Blain et al., 2010 [42]	1548 (100)	79 (77–81)												
Sabia et al., 2014 [62]	4016 (?)	73 (65–85)												
Georgiopoulou et al., 2016 [49]	2935 (52)	74 (70–79)												
Idland et al., 2013 [51]	113 (100)	79 (75–92)												
Camargo et al., 2016 [45]	2176 (54)	63 (35–84)												
Looker 2015 [53]	2975 (49)	≥50												
Cooper et al., 2014 [16]	2766 (51)	53												
Bravell et al., 2017 [92]	585 (59)	67 (60–91)												
Ojagbemi et al., 2015 [59]	2179 (?)	≥65												
Stijntjes et al., 2017 [65]	2979 (55)	72 (55–90)							* 					
Osuka et al., 2020 [60]	725 (62)	77 (73–80)												
Tian et al., 2019 [66]	201 (46)	79												
Buracchio et al., 2010 [44]	204 (58)	79 (≥65)												
Sakurai et al., 2017 [33]	223 (48)	73 (65–85)												
Bullain et al., 2016 [81]	578 (70)	93 (≥90)												
Doi et al., 2019 [88]	4086 (52)	72 (≥65)												
Donoghue et al., 2017 [89]	2250 (52)	72 (65–98)												
Briggs et al., 2019 [43]	3615 (?)	63 (≥50)												
Veronese et al., 2017 [67]	1732 (28)	69 (≥50)												
Veronese et al., 2017b [68]	970 (55)	72 (65–96)								* 				
Sanders et al., 2012 [63]	1459 (47)	69 (≥65)								* 				
Davis et al., 2015 [86]	308 (63)	≥70												

N = sample size; TUG = Timed Up&Go test. “Other” includes= maximum step length, speed of movement and stair mounting test. CVD = Cardiovascular Disease. * Partial predictive values due to sex or age range (see Appendix A). 

 = significant association; 

 = not associated or non-significant association; 
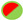
 = association found only in some outcomes. ? = female and male sample size are presented together.

A recap of the different cut-off points reported for gait speed and the TUG test and the risk of adverse health outcomes is shown in Appendix A. For gait speed, a cut-off point <0.8 m/s was associated with the higher number of health outcomes. However, for the TUG test, different cut-off points were proposed.

The predictive validity of the different motor fitness and flexibility tests for diverse health outcomes are detailed in Appendix A. The main findings regarding the levels of evidence and their predictive validity are presented below:

### 3.4. Predictive Validity for Falls and Fall-Related Outcomes

A total of 25 studies [6,7,8,9,31,34,35,36,37,54,64,70,72,73,74,75,76,77,78,79,82,83,84,85,87] and 2 systematic reviews [14,15] included falls or hip fracture as health outcomes (Table 3 and Appendix A).

#### 3.4.1. Gait Speed Tests

Gait speed was reported in 13 studies [6,7,8,9,31,34,35,36,37,54,64,70,82] and 2 systematic reviews [14,15], the 6 m gait speed test being the most common. One study also included step length to assess gait variables [54].

There was strong evidence for slower gait predicting falls in adults over 60 years old, as a result of seven selected studies [6,7,8,9,35,38,64] plus seven studies derived from two systematic reviews [14,15]. Three studies did not found association between gait speed tests and falls [31,36,82]. There was limited evidence for a short step length (<0.45 m), assessed over 5m predicting risk of falling in older adults over 70 years [54]. 

There was moderate evidence indicating that slower gait speed predicts hip fracture in older women over 75 years [34,70] and limited evidence, due to a low number of studies, in adults over 55 years old [37]. 

#### 3.4.2. Postural Balance Tests

Balance was assessed in 13 studies [54,70,72,73,74,75,76,77,78,79,82,83,84] and a systematic review [14]. Different protocols were used, such as the Berg balance scale [74,77], the Tinetti scale [54], functional reach test [14], tandem and semi-tandem stance or one-leg stance [70,73,78,82,84], with some of these tests performed in a force platform [72,75,76,79,83]. 

There was strong evidence reporting that impaired balance (worse performance or inability to complete the test) predicts falls in adults over 40 years old as a result of 11 studies [54,72,73,74,75,76,77,79,82,83,84]. A systematic review (with three studies) [14] and one study [78] did not found association between balance and falls.

There was limited evidence indicating that impaired balance predicts hip fracture in women over 75 years old [70].

#### 3.4.3. Multidimensional Measurement Tests

The TUG is a multidimensional test that measures mobility skills, combining gait speed, balance, and functional capacity [94].

The TUG was reported in 10 studies [31,35,36,73,74,78,82,83,85,87] and a systematic review [14].

There was strong evidence for worse TUG performance predicting falls (and fear of falling) in adults over 40 years old as a result of nine studies [31,35,36,73,74,82,83,85,87]. A systematic review (with three studies) [14] and a study did not find association between TUG and falls [78].

#### 3.4.4. Flexibility Tests

There was limited evidence, due to a limited number of studies (only one), that reduced hip and knee range of motion were associated with falls in older adults over 70 years [54].

### 3.5. Predictive Validity for Cognitive Decline and Impairment

A total of 11 studies [33,44,50,59,60,64,65,66,81,88,89], 2 systematic reviews [15,26], and 2 meta-analysis [24,25] included cognitive health outcomes (Table 3 and Appendix A).

#### 3.5.1. Gait Speed Tests

Gait speed was reported in eight studies [33,44,50,59,60,65,66,89], and four included systematic reviews/meta-analysis [15,24,25,26], where different distances were employed.

There was strong evidence for slower gait speed predicting cognitive decline and impairment (including develop of dementia and Alzheimer’s disease) in adults over 55 years old [15,24,25,26,44,50,59,65,66]. However, three selected studies found no association between gait speed tests and memory complaints [33] or cognitive decline [60,64].

#### 3.5.2. Postural Balance Tests

One study reported balance assessment, determining limited evidence indicating that impaired standing balance predicts incident dementia in adults over 90 years old [81].

#### 3.5.3. Multidimensional Measurements

Due to inconsistent results found in high-quality studies [88,89], there was limited evidence indicating that the TUG test predicts cognitive decline and impairment.

### 3.6. Predictive Validity for Depressive Symptoms and Well-Being

A total of four studies [43,63,67,68] included depressive symptoms and one well-being as health outcomes [86] (Table 3 and Appendix A).

#### 3.6.1. Gait Speed Tests

There was moderate evidence indicating that slower gait speed predicts depressive symptoms in adults over 50 years old [43,67] and in older men over 65 years [63,68].

#### 3.6.2. Multidimensional Measurement Tests

There was limited evidence indicating that worse TUG test performance predicts decline in well-being in older adults over 70 years [86].

### 3.7. Predictive Validity for Mobility Limitations, Disability and Frailty

A total of 16 studies [11,32,38,39,40,41,46,47,52,55,56,61,69,80,90,91], 2 systematic reviews [14,15], and a meta-analysis [27] included related-mobility disability and frailty as health outcomes (Table 3 and Appendix A).

#### 3.7.1. Gait Speed Tests

Gait speed was explored in 13 selected studies [11,32,38,39,40,41,46,47,52,55,56,61,80] and 3 reviews [14,15,27], where different distances were employed.

There was strong evidence for slower gait speed that predicts disability in instrumental activities of daily living (IADL) in adults over 54 years old, as a result of seven selected studies [11,38,39,40,41,55,56] and three reviews (composed of 61 studies) [14,15,27].

There was strong evidence indicating that slower gait speed predicts mobility disability in adults over 50 years old, as a result of four selected studies [32,46,47,61] and two systematic reviews (composed of 10 and 3 studies, respectively) [14,15].

There was limited evidence indicating that slower gait speed predicts frailty status in older adults over 65 years [52], and that faster gait speed predicts greater functional autonomy in older adults over 68 years, especially in women [80].

#### 3.7.2. Postural Balance Tests

Balance was assessed in three studies [55,56,80], a systematic review [14] and a meta-analysis [27]. Different protocols were used to assess balance.

There was strong evidence indicating that impaired balance predicts disability in IADL or mobility disability in adults over 40 years old [14,27,56]. Makizako et al. [55] did not find balance predictive of disability in IADL in older adults over 68 years.

There was limited evidence indicating that balance could moderately predict functional decline in older adults over 68 years, especially in women [80].

#### 3.7.3. Multidimensional Measurement Tests

The TUG test was reported in two studies [80,90] and two reviews [14,27].

Due to inconclusive results, limited evidence was found for TUG performance and disability in IADL or mobility disability in older adults. A meta-analysis [27] found it predictive, but a systematic review did not [14].

Due to a limited number of studies (only one per health outcome), there was limited evidence indicating that worse TUG performance identifies frailty status [90] and loss of functional autonomy in older adults over 65 years [80].

#### 3.7.4. Flexibility Tests

There was limited evidence indicating that poor flexibility predicts mobility disability in older adults 65 years [69].

#### 3.7.5. Other Tests

Three studies included other tests [41,69,91] to assess the predictive validity on mobility limitations, disability, and frailty.

Due to a limited number of studies (only one per each test), limited evidence was found. Ward et al. [69] reported that speed of movement test predicts mobility disability mobility disability in older adults over 65 years. Adachi et al. [41] employed a maximum step length test to assess disability in IADL, and found that the test predicts disability in IADL in older adults over 76 years. Finally, Schroll et al. [91] found that the stair mounting test predicts disability in IADL in older men over 75 years.

### 3.8. Predictive Validity for Cardiovascular Disease Risk and Mortality

A total of 19 studies [3,11,12,16,40,42,45,48,49,50,51,52,53,57,58,62,64,71,92] and a systematic review [15] included CVD risk, stroke, all-cause mortality, and other-causes mortality as health outcomes (Table 3 and Appendix A).

#### 3.8.1. Gait Speed Tests

Gait speed was reported in 17 selected studies [3,11,12,32,37,44,53,56,57,58,59,64,72,74,78,80,92] and a systematic review [15]. The 2.4 m, 4 m, and 6 m gait speed were the most employed tests.

There was strong evidence indicating that slower gait speed predicted CVD risk in adults over 45 years old [11,49,57,58]. However, Lee et al. [12] found no interaction between gait speed and CVD in older adults.

There was strong evidence for slower gait speed that predicts all-cause mortality in adults over 35 years old [3,15,40,42,48,49,50,51,52,57,58,62,64,71]. Lee et al. [12] found not interaction between gait speed and all-cause mortality in older adults.

There was moderate evidence for slower gait speed that predicts other-causes mortality in adults over 50 years old [53,58].

Regarding stroke, there was limited evidence indicating that slower gait speed was not associated with stroke risk in adults over 35 years old [45].

#### 3.8.2. Postural Balance Tests

Balance was reported in four studies, [16,42,51,58] where different protocols were used.

There was strong evidence indicating that impaired balance predicts all-cause mortality in adults over 53 years old [16,42,51,58].

There was limited evidence indicating that impaired balance predicts CVD risk in older adults over 65 years [58]. Moreover, there was limited evidence for impaired balance predicting other-causes mortality in older adults over 65 years [58].

#### 3.8.3. Flexibility Tests

There was limited evidence indicating that the touch-toes test does not predict all-cause mortality in adults over 60 years old [92].

### 3.9. Predictive Validity for Institutionalization or Hospitalization

One study [71] and two systematic reviews [14,15], composed of two and five studies, respectively, included institutionalization or unplanned hospitalization as health outcomes (Table 3 and Appendix A).

#### 3.9.1. Gait Speed Tests

Gait speed was reported in the selected study [71] and the systematic reviews [14,15], where the 6m gait speed is the most common test.

There was strong evidence indicating that slower gait speed predicts risk of institutionalization or hospitalization in adults over 60 years old [14,15,71].

#### 3.9.2. Postural Balance Tests

There was limited evidence indicating that impaired balance predicts risk of hospitalization in adults over 60 years old [14].

## 4. Discussion

The purpose of this study was to systematically examine the predictive validity of existing motor fitness and flexibility tests on health outcomes in adults and older adults.

Results from this systematic review revealed that there exists strong evidence indicating that: (1) slower gait speed predicts falls in adults over 60 years old, cognitive decline and impairment (including develop of dementia and Alzheimer’s disease) in adults over 55 years old, mobility disability in adults over 50 years old, disability in IADL in adults over 54 years old, CVD risk in adults over 45 years old, all-cause mortality in adults over 35 years old, and risk of institutionalization or hospitalization in adults over 60 years old; (2) impaired balance predicts falls in adults over 40 years old, disability in IADL or mobility disability in adults over 40 years old, and all-cause mortality in adults over 53 years old; (3) worse TUG performance predicts falls and fear of falling in adults over 40 years old.

We have also found that there exists moderate evidence indicating that: (1) slower gait speed predicts hip fracture in older women over 75 years, depressive symptoms in adults over 50 and older men over 65 years, and other-cause mortality in adults over 50 years old; (2) worse TUG performance predicts falls in women over 40 years old.

Due to a limited number of studies, the results also suggested limited evidence showing that: (1) slower gait speed predicts hip fracture in adults over 55 years old, frailty status in older adults over 65 years, decrease in functional autonomy in older adults over 68 years, and that does not predict stroke risk in adults over 35 years old; (2) impaired balance predicts hip fracture in women over 75 years old, incident of dementia in adults over 90 years old, functional decline in older adults over 68 years, CVD risk in older adults over 65 years, other-cause mortality in older adults over 65 years, and risk of hospitalization in adults over 60 years old; (3) worse TUG performance predicts cognitive decline and dementia in older adults over 65 years, disability in IADL in older adults over 65 years, decline in well-being in older adults over 70 years, and frailty status and loss of functional autonomy in older adults over 65 years; (4) poor flexibility predicts falls in older adults over 70 years, mobility disability in older adults 65 years, and does not predict all-cause mortality in adults over 60 years old; (5) a shorter step length predicts risk of falling in older adults over 70 years, slower speed of movement predicts mobility disability in older adults over 65 years, shorter maximum step length predicts disability in IADL in older adults over 76 years, and worse performance in the stair mounting test predicts disability in IADL in older men over 75 years.

At the age of 30, the biological system functioning reflects a critical transition point into decline and ageing [57]. Adults over 60 years old suffer from more diseases related to aging [1], and they are users of more healthcare resources [71]. Ageing is characterized by gait changes, which may be a manifestation of compromised motor executive function [51]; then, these results imply the neurological ability to coordinate motor tasks, reflecting in motor fitness tests, such as gait speed, balance, or the TUG tests. 

The results derived from the studies included in the current systematic review suggest a predictive value of motor fitness tests related to diverse adverse health outcomes mainly for adults aged over 60, which is in agreement with this fact. Indeed, previous reviews are in accordance with our findings in this population [14,15,24,25,26,27]. 

Moreover, predictive value has also been found for motor fitness tests related to health outcomes in younger adults. Thus, an early screening of physical fitness performance could help prevent adults from more harmful aging deterioration. For instance, evidence extracted from the present review seems to indicate adverse health outcomes from 35 years old related to slower gait speed (mobility disability, disability in IADL, CVD risk, all-cause mortality), or related to impaired balance or worse TUG performance (i.e., risk of falling, disability). Therefore, knowing the predictive ability of each test to detect health outcomes can facilitate to incorporate an early screening into clinical and practical settings.

### 4.1. Predictive Validity of Gait Speed Tests

Walking ability is a global measure of mobility that reflects a basic aspect of daily activity [40]. Gait speed seems to begin to decrease in old age [26] and is related to difficult in walking, which may indicate more healthcare needs, higher incident disability, and shorter life expectancy [40]. Our review supports the idea that gait speed assessment might be used for risk stratification and to guide health practitioners and clinicians in the management of altered health outcomes.

Although physical assessment is proposed mainly for older adults [13], and all the selected reviews corroborated it, as they included only older adults [14,15,24,25,26,27], based on the present review, we propose the inclusion of younger populations. Specifically, gait speed assessment is a good tool to identify adults at risk of cognitive decline and impairment, mobility disability/IADL, CVD and all-cause mortality in adults from 35 years old. Moreover, gait speed may also identify depressive symptoms in adults over 50 years old. However, further research is needed, because only two studies have explored this relationship [43,68].

The elderly population shows a wide range of adverse health outcomes, including those mentioned above. In this sense, assessment of gait speed is currently recommended for older adults in the health practice environment [7,76]. Gait speed is a good measure of overall ability to compensate for decline in multiple body systems, including sensorimotor and cognitive function, which are common risk factors for falls [7]. The reduction in gait speed may be due to loss of physical functioning or the deterioration of brain motor control centers [8,26]. Moreover, evidence suggests that inflammatory markers [26,48] play a role in the association between gait speed and mortality events, since they are associated with disability, worse cognitive performance and motor functioning, frailty, and death [26,48].

Different gait speed cut-off points have been reported for risk stratification of adverse health outcomes, but a single threshold was not yet evident. Nevertheless, and independent to the distance of the chosen test, cut-off points of <0.8 m/s and <1.0 m/s in gait speed appear to be sensitive to predict the risk of most of these adverse health outcomes in adults and older adults (Appendix A). In fact, these cut-off points have been previously proposed [15,26].

### 4.2. Predictive Validity of Balance Tests

Maintaining balance is a complex task that demands good functioning of multiple organ systems with an accurate coordination between them [64]. The impaired balance may be due to lack of strength or age-related deterioration of sensory and neuromuscular control mechanisms [75,76].

Although gait speed assessment is well-established, strong evidence indicates that impaired balance is a good tool to identify adults over 40 years old at risk of falling, disability in IADL or mobility disability, and all-cause mortality in adults over 53 years old.

Limited evidence was found for balance and other health outcomes in older adults, such as hip fracture, incident of dementia, functional decline, CVD risk, and other-cause mortality risk. This limited evidence may be primarily due to a lack of consensus on which balance test is most accurate in measuring different health outcomes or populations. A recent meta-analysis has suggested that the one-leg balance test is a good predictor of disability in IADL in older adults [27], although only three studies pooled the analysis. In fact, establishing cut-off points for balance measurement is also challenging, due to the wide range of existing protocols, although some studies have identified that inability to stand on one leg for less than 5 s may be sensitive to changes in risk of falls [74,82,84]. Therefore, more studies are still needed to clarify which protocol is the most accurate in order to identify the most sensitive cut-off in balance performance related to different health outcomes.

### 4.3. Predictive Validity of Multidimensional Measurement Tests

The TUG test was a reliable and valid test developed by Podsiadlo and Richardson [94] in 1991 to assess “basic mobility skills” in older adults (70–84 years). This test was intended to be a simple and useful compendium between the measures of gait speed, balance, and functional capacity [94]. Although various factors are associated with falls, mobility problems and impaired balance have been consistently identified as the main risk factors [73,74,78,83].

The TUG test has been identified as a good tool to identify adults over 40 years old at risk of falling and fear of falling [31,35,36,82,85,87]. 

Moreover, worse TUG performance may also predict falls in women over 40 years old [73,74,83].

Therefore, the TUG test could be proposed as a predictive test for risk of falling in community dwelling adults from 40 years old [31,35,36,73,74,82,83,85,87]. Since the risk of falling is associated with loss of independence, injuries, disability, long-term health care and premature mortality [31,73,74,83], an available predictive tool for risk of falling is advisable. The adverse health outcomes related to health have been especially identified for women [73,83], due to the loss of estrogens derived from perimenopause (around the age of 40) that can affect bone quality, being more prone to bone deterioration and fractures (such as hip fracture) [95].

Hence, falls and their consequences could be prevented with early screening by evaluating the TUG test among the aging adult population.

Nevertheless, this motor fitness test presents a series of drawbacks, resulting in limited and inconclusive predictive value in relation to other health outcomes.

Since the TUG test combines gait speed plus balance skills, it is not surprising that in those studies where some of these skills have been evaluated simultaneously (TUG performance plus gait speed or balance performance), all of them proved predictive capacity. In fact, the TUG test, gait speed and balance test were related to falls [54] and to decline in functional autonomy [80] in older adults; the TUG and a gait speed tests were related to falls in older adults [35]; or the TUG and balance tests were related to falls in adult women over 40 years old [74,83].

Conversely, Mulasso et al. [78] found that neither the TUG nor a one-leg standing balance tests were predictive of falls. The same results were derived from the systematic review conducted by Cavanaugh et al. [14] between the TUG and the functional reach tests (a balance test) related to falls and mobility disability. Nevertheless, only a few participants reported experiencing at least one fall in the previous 12 months, 30 [78] and 42 [14] participants, respectively, which could imply such a lack of significance.

Donoghue et al. [89] also failed to find any association between the TUG and gait speed tests related to cognitive decline. However, trends of significance were found between slower TUG performance and increased cognitive decline. They also reported that their participants had a mean of 10.5 s in TUG performance and their cognitive performance did not change substantially over the follow-up. Thus, that could partially explain its lack of predictive validity.

On the other hand, these simultaneous results have not always been given among tests. Thus, some studies found predictive value of the TUG test and failed to find association regarding falls risk for gait speed tests [31,36,82]. One possible explanation for that fact could be the minimum difference in gait speed tests performance between fallers and non-fallers, since it difference was only of 0.05 m/s [31], 0.12 m/s [36], and 0.13 m/s [82], respectively, between groups, its difference being higher in TUG performance (for fallers vs. non-fallers).

The fact that the health outcome “falls” had the highest number of studies measured with the TUG test, gait speed, and balance tests measured simultaneously may be partly due to that, precisely, avoiding falls require of good mobility functions, such as walking ability, maintaining balance, strength, and agility [31].

Inconclusive evidence was found for the predictive validity of the TUG test related to IADL. In the meta-analysis conducted by Wang et al. [27] they suggested that the TUG test is a good tool to identify disability in IADL. However, Cavanaugh et al. [14] found insignificant results. This fact could be partially explained by the number of studies explored in each review. While Wang et al. [27] included eight studies analyzing TUG, Cavanaugh et al. [14] only included two, which may limit the predictive evidence found for such test.

Cut-off points in TUG performance are still unclear. Therefore, different values have been previously reported. For example, cut-off points of 8.1, 9.2, and 11.3 s for adults aged 60–69, 70–79, and 80–99 years old, respectively, has been described by Bohannon [96] in 2006 to identify elders with mobility deficits. We have summarized the cut-off points found in TUG to identify diverse health outcomes, such as falls, functional autonomy decline, disability in IADL, frailty, or well-being decline, ranging from faster performance (7 s) to slower (21 s) (Appendix A).

### 4.4. Predictive Validity of Flexibility Tests

Although several cross-sectional studies have found associations between flexibility and diverse health outcomes, such as cardiometabolic risk, well-being, etc., its predictive value (through a longitudinal design) is still scarce [54,69,92]. These three studies were carried out in older adults with different flexibility protocols. Furthermore, only Ward et al. [69] found that flexibility was predictive of mobility disability [69], while the other two studies did not confirm its predictive capacity for falls [54] or all-cause mortality [92].

### 4.5. Limitations and Strengths

This systematic review has some limitations that should be considered when interpreting the results. First, although our intention was to provide a representative sample in adults and older adults, the reality was the impossibility of finding a sample of adults under 35 years of age. This fact may be explained by the fact that ageing deterioration is the main reason of multiple adverse health conditions such as falls, frailty, cognitive decline, etc. [1]. Therefore, the clinical screening of that younger population is not yet set. Second, the results of the present systematic review should be interpreted with caution due to the variety of tests used to assess motor fitness and flexibility, the heterogeneity in statistics reporting risk prediction, and protocols that make difficult to directly compare results across studies and that could somehow partially change the results presented here. Likewise, heterogeneity among follow-up lengths limited direct comparison between studies as different follow-up timeframes were employed (e.g., 1 year vs. 25 years). Similarly, another limitation is the inconsistent utilization of cut-off scores for the baseline measures (in both, motor fitness and flexibility tests and health outcomes) across selected studies. Finally, we have only included apparently healthy population at baseline, so it is possible that some motor fitness and flexibility tests may have predictive value for some specific pathologies or particular health conditions.

Finally, several strengths of this systematic review deserve to be highlighted. The major strength of our study is the wide range of health outcomes introduced in the research process, including the biological and psychological spheres, providing a comprehensive overview of health, in adults and older adults. In this sense, another important contribution is the inclusion of those health outcomes with scarce literature, such as fear of falling, frailty, unplanned hospitalization, depression, or well-being. Moreover, it includes data from near 375,300 apparently healthy adults and older adults, with a mean follow-up greater than 8 years.

### 4.6. Perspective

Poor physical performance on relatively simple tasks may indicate a deterioration in health. Therefore, it is a major public health concern to bridge the gap between life expectancy and healthy life expectancy, so that people can live old age without reduced QoL.

The expected objective of these findings is to help identify contributing factors and design strategies to minimize the risk of developing adverse health outcomes and improve the autonomy/QoL of adults and older adults through a simple scan of their motor fitness and flexibility status. This systematic review supports previous finding of screening of physical fitness as a powerful marker of health in adults and older adults.

Motor fitness and flexibility tests are quick, simple, and inexpensive tools that can be administered with minimal training and equipment in research and clinical settings. Under evaluation time constraints in clinical settings, gait speed assessment remains the main motor fitness test recommended to identify a multidimensional approach, such as age-related health deterioration.

## 5. Conclusions

This systematic review emphasized important major points regarding the predictive validity of motor fitness tests in adults and older adults.

Slower gait speed predicts falls, cognitive decline and impairment (including develop of dementia and Alzheimer’s disease), mobility disability, disability in IADL, CVD risk, all-cause mortality, and risk of institutionalization or hospitalization. Its use is suggested especially in adults over 35 years old.

Impaired balance predicts falls, disability in IADL or mobility disability, and all-cause mortality. Its use is suggested especially in adults over 40 years old.

Worse TUG performance predicts falls and fear of falling in adults over 40 years old, especially in women.

Therefore, these results provide further justification to integrate the measurement of motor fitness, mainly gait speed and balance, in the prognostic assessment tools as indicators of health outcomes (such as falls risk, cognitive impairment, disability, CVD risk, all-cause mortality, and hospitalization) in both adults and older adults. Identifying these risk factors will allow for early intervention based on adequate levels of physical fitness and potentially decrease morbidity and mortality over lifetime, at the same time increasing their QoL.

Finally, in spite of the limited evidence regarding flexibility, our results suggest that flexibility testing may help detect older adults at risk of falling, suffering from disability, or expecting all-cause mortality. Therefore, future studies focusing on the predictive value of flexibility tests in this population are warrantied. Furthermore, it is still necessary to elucidate the relationship between those motor fitness tests that present limited evidence in relation to the predictive value for various health outcomes.

## Figures and Tables

**Figure 1 jcm-11-00328-f001:**
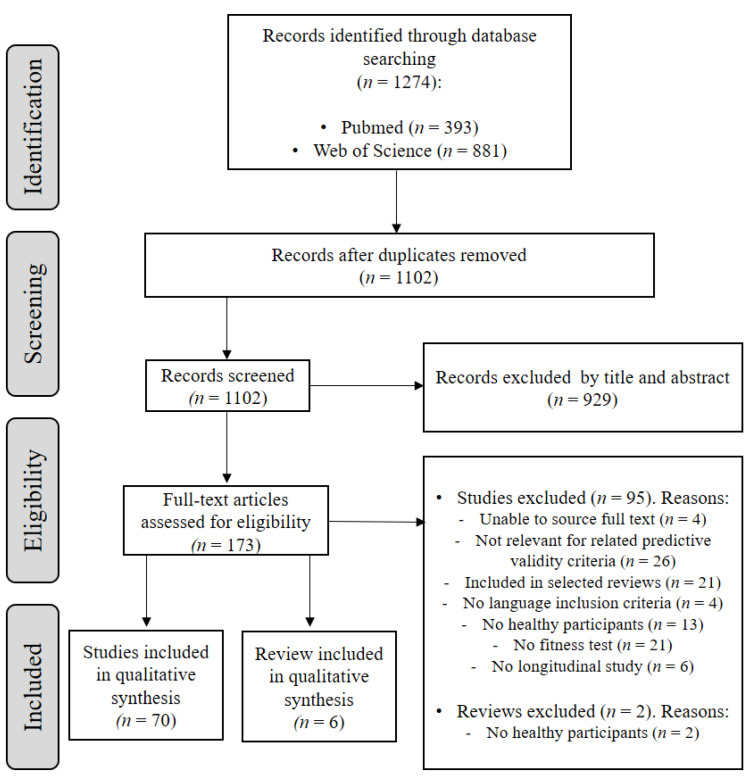
Flow chart of the study selection process.

## Data Availability

The authors agree to share their raw data and any digital study materials as appropriate.

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
