# Peer review of "Predictive Validity of Motor Fitness and Flexibility Tests in Adults and Older Adults: A Systematic Review"

_jcm, 2022, doi:10.3390/jcm11020328_

Round 1

Reviewer 1 Report

I think the study is relevant. But it is necessary to make a greater synthesis of results since these are very extensive and are not well understood. Put more selective inclusion criteria and obtain new results.

Author Response

Dear Reviewer, please see the attachment.

Best regards,

APB

Reviewer 2 Report

The review performed is comprehensive and conclusions drawn are well supported by the review and discussion. Few comments that have to be addressed are as follows:
  • Could further discussion/review and clarification be included in the discussion section on validity of TUG test for fall-risk assessment?
  • Kindly proof read the manuscript. There are quite a few grammatical errors and typos.

Author Response

Dear Reviewer, please see the attachment.

Best regards,

APB.
